# The Effect of the Height of Coppicing and Harvest Season on the Yield and Quality of the Essential Oil of *Kunzea ambigua*

**DOI:** 10.3390/plants12010020

**Published:** 2022-12-20

**Authors:** Chanjoo Park, Sandra M. Garland, Dugald C. Close

**Affiliations:** Horticulture Centre, Tasmanian Institute of Agriculture, University of Tasmania, Hobart 7001, Australia

**Keywords:** kunzea oil, defoliation, harvest intensity, harvest season, plant defense

## Abstract

*Kunzea ambigua* is a small shrub belonging to the *Myrtaceae* family and the leaves are steam-distilled to produce a therapeutically active essential oil. With production moving from wild-harvested to orchardised stands, there is a need for harvest management of kunzea oil. This study compared the regrowth, essential oil content and composition of *kunzea* plants after harvesting vegetative material to a depth of 0.2 m above ground level (shallow-cut), relative to plants cut to a depth of 0.1 m above ground level (deep-cut) over the 2018/2019 growing season. Increased vegetative biomass accounted for the increased oil yield and was caused by consistently higher growth rates of 50 to 60% across all seasons in shallow-cut crops relative to those subject to deep-cut. Total soluble sugar concentrations were higher in the leaves and lower in the roots of deep-cut treated plants compared to the other treatments, indicating defoliated *K. ambigua* responds by mobilising sugars into above-ground biomass. The overall essential oil content of leaves was constant regardless of season, though the oil yield for shallow-cut was 1.9-fold higher at 11.79 ± 0.23 g/m^2^ compared to deep-cut (6.24 ± 0.18 g/m^2^). An interactive effect of harvest intensity with season was recorded for all major components except for a non-significant effect of season on terpinen-4-ol. Bicyclogermacrene and α-pinene were elevated in both shallow- and deep-cut treatments relative to control (un-cut) in spring, possibly due to the plant defense response after de-foliation. The highest percentage of bioactive compounds (1,8-cineole and viridiflorol) were present in autumn. Therefore, the recovery of biomass post-harvest is optimised by shallow-cut harvests, and the profile of kunzea oil can be manipulated to elevate levels of specific bioactive components by selecting to crop in autumn/spring.

## 1. Introduction

*Kunzea ambigua* is a woodland shrub belonging to the *Myrtaceae* family and is native to south-eastern Australia and Tasmania [1,2]. It grows naturally to a height of 1.5 m and spreads to approximately four meters [2]. The fine, lanceolate leaves of *K. ambigua* aggregate to form dense foliage that contains strongly aromatic and volatile compounds, which are harvested and steam-distilled to produce a pale-yellow essential oil. Kunzea oil has a spicy, pine-eucalyptus odour and it has been widely used as an insect repellent and as a healing agent for a wide range of ailments by the first nations peoples of Australia [3,4,5]. Currently, commercial products with kunzea oil have many uses in cosmetics and personal care products such as oil balm, body spray and soap. The economic viability and expansion of plantation production are closely related to systematic cultivation and management strategies to allow for the steady supply of quality plant materials for essential oils production [6,7]. When repeatedly coppiced, it forms a sprawling growth habit, and a broad ground cover crop is achieved following multiple harvests. However, there have been no studies to establish optimal harvest intensities and the effect of season on the yield and chemical profile of kunzea oil.

Plants have several compensatory evolved mechanisms to recover following defoliation [8]. Generally, partial plant defoliation induces up-regulation of photosynthesis in the remainder of the canopy and non-structural carbohydrates (NSCs) are redirected to provide energy for the regrowth of shoots [9]. Levels of NSCs are a key indicator of plant growth and adaptation strategies [10,11]. Sucrose is the major form of organic carbon exported from the photosynthetic sources and reserves to sink organs, and this process is essential for the survival and productivity of plants through the provision of energy for respiration during the early stages of leaf growth that drives rapid re-foliation [12,13,14]. In the case of the commercial production of tea (*Camellia sinensis* L.), regular partial defoliation stimulated new shoot growth [15,16]. When defoliation was too severe, carbohydrate reserves were progressively depleted, resulting in the death of roots and the failure of new buds to form shoots [17]. Defoliation can also trigger plant defence mechanisms, including changing the chemical profile because of the upregulation of different genes [18]. Terpenoids are important for plant survival and are produced in response to biotic and abiotic stress [19]. The production of sesquiterpenes, such as bicyclogermacrene, is associated with a defence response against pathogens and herbivores in members of the family *Solanaceae*, *Vitaceae*, and *Meliaceae* [20]. Indeed, abiotic stress can be managed as an agronomic strategy for improving the quality of horticultural produce through enhanced levels of bioactive constituents such as bicyclogermacrene and α-pinene [21].

The seasonal timing of harvest of essential oil crops is critical as it can directly affect the oil yield and composition, though this has been shown to be species-specific and can be further confounded by the combined interactions of plant growth and development ontogeny and abiotic factors [22,23,24]. Hussain et al. [25] found that the yield of basil oil (*Ocimum basilicum* L.) ranged from 0.5 to 0.8% DW across seasons with maximums recorded in winter, whereas the lower oil content present in summer vegetation may have bene caused by evaporation. Similarly, the total oil content of peppermint (*Mentha piperita* L.) increased up until flowering in early summer, then slowly declined, although the bioactive component of menthol continued to increase [26]. Furthermore, seasonal variation in the levels of terpinen-4-ol in tea tree oil (*Melaleuca alternifolia*) has been reported [27]. Therefore, the study aims to determine the influence of harvest intensity (control, shallow- or deep-cut) and season on biomass accumulation, oil yield and composition of kunzea oil. Furthermore, the level of NSCs in the leaves, stems and roots were measured to show how *K. ambigua* mobilises resources after partial defoliation caused by harvest.

## 2. Results

### 2.1. Growth Rates and Above-Ground Biomass following Harvest of Kunzea Plants According to Harvest Intensity and Season

To understand plant responses to different levels of harvest intensity, cumulative above-ground biomass and GR of biomass was investigated post-harvest. For cumulative above-ground biomass (Figure 1), control and shallow-cut treatments (947.42 ± 20.87 and 804.52 ± 90.44 g DW/m^2^, respectively) were nearly two times higher than deep-cut treatment (404.91 ± 62.59 DW/m^2^) at the end of spring. This was reflected in the GR which was significantly higher in control and shallow-cut, relative to the deep-cut treatment in late summer and autumn. In relation to seasonal variation, the GR of the control was three times higher in late summer (6.98 ± 0.54 g DW/m^2^/day) than it was in the winter/spring (2.76 ± 0.16 g DW/m^2^/day). Significant interactions between harvest intensity and season on cumulative above-ground biomass (*p* < 0.05) and GR of biomass (*p* < 0.0001) were found (Table 1).

### 2.2. The Levels of Starch, Soluble Sugars and NSCs in Coppiced Kunzea Plants Influence on Essential Oils Biosynthesis

There were significant differences in the starch levels between harvest intensity treatments (Table 2), whereby levels were highest in both the leaves and the roots of the control (2.51 ± 0.08 mg/g and 9.31 ± 0.89 mg/g respectively) and decreased in harvested plants with the lowest levels detected in the leaves and roots of deep-cut treatments of 1.82 ± 0.04 mg/g and 7.51 ± 0.59 mg/g, respectively. Conversely, TSS was highest in the leaves of the deep-cut treatment (73.25 ± 3.71 mg/g) and lowest in the roots at 18.41 ± 0.77 mg/g, relative to the control (24.23 ± 3.45 mg/g). The sugar most prevalent in the leaves of shallow- and deep-cut treatments was sucrose (41.54 ± 2.22 mg/g and 44.96 ± 2.17 mg/g, respectively), which was significantly higher compared with the control treatment (23.53 ± 0.46 mg/g). Levels of sucrose in the roots showed the opposite pattern. Overall, the concentration of NSCs in the leaves of the deep-cut treatment was the highest (75.07 ± 3.74 mg/g) whereas that in roots was the lowest (25.92 ± 0.96 mg/g).

As shown in Table 3, essential oil content in the deep-cut treatment (1.54–1.66% DW) was 17% lower than the levels distilled from the leaves from control and shallow-cut treatments (1.80–1.90% and 1.84–2.00% DW, respectively). There was no seasonal variation in essential oil content as a percentage of biomass, however, significant interaction (*p* < 0.001) was found between harvest intensity and season on the overall essential oil yield (g DW per m^2^). In spring, the essential oil yield in shallow-cut (11.79 ± 0.23 g DW/m^2^) and deep-cut (6.24 ± 0.18 g DW/m^2^) treatments were lower by 30.85% and 64.40%, respectively, of that recovered from the control (17.05 ± 1.21 g DW/m^2^). The control, however, had accumulated 7.01 ± 1.01 g DW/m^2^ prior to the start of the trial (time zero), and the new growth provided for an increase in the essential oil yield of approximately 10.04 g DW/m^2^, lower than that accumulated in those that had been harvested by shallow-cut.

The trial site had been commercially harvested in April 2018, so that there had been eight months of growth prior to the start of this trial. When considered in terms of actual commercial operations, the plantation owner had only one harvest from the control (un-cut) from April 2018 to December 2019 but had two harvests from the two harvest intensity treatments. Therefore, to compare the overall oil yield (g DW/m^2^) of the control (one harvest) to that of areas subject to two harvests, the oil collected from the leaves from shallow- and deep-cut treatments that had accumulated prior to December 2018 was added to the oil yield accumulated from December 2018 to December 2019 and, in this study, is referred to as the cumulative essential oil yield. There was a significant difference in cumulative essential oil in spring between control (17.05 ± 1.21 g DW/m^2^) and shallow-cut treatment (14.90 ± 0.48 g DW/m^2^), whereas the deep-cut treatment (13.25 ± 0.65 g DW/m^2^) was significantly lower by nearly 22%, relative to control. In particular, vegetative growth in control was 20 months yet those of the shallow- and deep-cut treatments was less than 12 months from April 2018. Hence, the shallow-cut treatment together with two harvest times in a year would be conducive to increasing the oil quantity in a sustainable and profitable industry.

### 2.3. The Effect of Harvest Intensity and Season on the Chemical Constituents of Kunzea Essential Oil (%)

There were significant interactive effects of harvest intensity and season on α-pinene (*p* < 0.0001), globulol (*p* < 0.05), bicyclogermacrene (*p* < 0.05) and the minor components (*p* < 0.0001) (Table 4). In the control in spring, α-pinene (25.95 ± 1.98%) and bicyclogermacrene (3.94 ± 0.25%) were two times lower compared to shallow (42.56 ± 4.17 and 8.49 ± 0.06%, respectively) and deep-cut treatments (39.04 ± 1.77 and 7.58 ± 0.34%, respectively) (Figure 2A). In contrast, the percentage composition of minor components was highest in the control at the end of spring (39.07 ± 2.03%), which was nearly 1.5-fold higher than that recorded for shallow- and deep-cut treatments (27.16 ± 2.11 and 31.07 ± 1.11%, respectively). With respect to seasonal variation, oils produced from summer and autumn samplings had the highest percentage in overall bioactive compounds such as 1,8-cineole (range from 8.23 + 1.60 to 11.40 + 1.14%) and viridiflorol (range from 10.42 + 0.32 to 15.56 + 0.61%). The quality component, viridiflorol, decreased by 34% in spring. There was no seasonal variation in terpinen-4-ol (range from 0.28 + 0.03 to 0.49 + 0.04%) (Figure 2B).

Values are presented as means ± SD (*n* = 4). Mean values designated by a different letter are significantly different between groups (*p* < 0.0001, *p* < 0.001, *p* < 0.05) according to the two-way ANOVA test with initial harvest intensity (HI) and season (S) as variability factors: initial harvest intensity (HI) for a, b, c and season (S) for x, y and z using lowercase letters, and initial harvest intensity (HI) × season (S) interaction for capital letters. The zero-time data was not included in the two-way ANOVA test.

## 3. Discussion

### 3.1. Plant Growth and the Allocation of NSCs of K. ambigua in Response to Harvest Intensity and Season

Higher GR of biomass was recorded for the shallow-cut treatment relative to the deep-cut treatment in *K. ambigua* following harvest but was highest in the control (uncut) from the end of summer through to the end of spring. Moderate defoliation results in the emergence of new leaves with modified assimilatory capacity [28,29]. Quenti, et al. [30] reported that the removal of 45% of the leaves of blue gum (*Eucalyptus globulus* Labill.) was compensated for by an increased photosynthetic rate, improved water relations and increased utilization of carbon assimilates. However, heavily defoliated branches, such as those implemented in the deep-cut *kunzea* harvest, might result in there being insufficient resources within the remainder of the crown to recover and maintain re-growth [31]. It follows that the slow growth recorded immediately after deep-cut harvest in this study may have been caused by a lack of photosynthetic leaf area [30,32]. Under productive environments, plants maximise their growth rates by continuously generating new roots and leaves [33]. Conversely, in non-productive environments, where plant tissue is damaged, plants show reduced growth rate and instead mobilise reserves, allowing for re-growth [34]. This study quantified NSCs within the roots, leaves and branches of uncut, shallow-cut and deep-cut *K. ambigua* plants to better understand the allocation of resources in response to harvest intensity.

The defoliation from harvesting can alter carbohydrate metabolism. Monosaccharides are mobilized instead of sugar polymers and starch reserves are hydrolysed to soluble sugars [35]. In this study, the starch reserves in both roots and leaves were significantly lower in harvested *kunzea* plants relative to the control. Depleted starch reserves after deep-cut treatment are consistent with resources being mobilised for new shoot growth in response to partial defoliation from harvest. Mobilisation of reserves in response to increased harvest intensity was further demonstrated by significantly higher levels of TSS in the leaves, and lower levels in the roots of deep-cut treatments, relative to control and shallow-cut treatments indicating that, even after 12 months, the deep-cut treatment had still allocated resources to new shoot growth compared to other treatments. These findings are consistent with those reported in tea (*Camellia sinensis* L.) that has been shown to re-allocate NSCs from roots to shoots following partial defoliation, such that NSCs are highest in leaves after harvest [36]. Similarly, a decrease in starch content and a parallel increase of soluble sugar content was reported in balsam fir (*Abies balsamea* (L.) Mill) after it was subjected to heavy and moderate defoliation (61–80% and 41–60%, respectively), relative to the control treatment [37].

The sugar most prevalent in *kunzea* was sucrose, being higher in the leaves and branches and lower in the roots of harvested plants, compared to control. Likewise, fructose and glucose mirrored the trends observed for sucrose in leaves and branches, but not in the roots. This could indicate that new photosynthates might be preferentially allocated to organs such as new leaves and branches near the carbohydrate source to support growth [38]. Glucose plays a role as a substrate for cellular respiration or as an osmolyte to maintain cell homeostasis, whereas fructose seems related to secondary metabolite synthesis such as the production of phenolic compounds [39]. Nonetheless, there was consistency in the levels of fructose and glucose in roots of *K. ambigua* across all treatments, which may be required for the metabolic maintenance of roots [40].

### 3.2. The Effect of Harvest Intensity and Season on the Quantity of Kunzea Essential Oil

Slow growth is often accompanied by altered chemical characteristics such as elevated levels of defensive secondary compounds [41]. The results presented in this study reveal that the slower GR in defoliated *K. ambigua* may have also been due to the prioritisation of resources from re-growth to the production of defence compounds, such as bicyclogermacrene. Shallow-harvested *Kunzea* plants yielded more oil (% DW) than deep-harvested ones. Biotic and abiotic environmental factors affect plant growth, essential oil yield and chemical composition [42]. In lemongrass (*Cymbopogon flexuosus Stapf.*) only young, rapidly expanding leaves were metabolically active enough to synthesize essential oils when starch was mobilised. An adequate supply of carbon precursors, cofactors and energy is required to simultaneously produce essential oils and vegetative material post-harvest [12,43]. In this study, the levels of sucrose in the remaining leaves of cut *K. ambigua* were similar in the shallow- and deep-cut plants yet oil content was significantly higher in the shallow-cut treatment, indicating that there were insufficient resources for both secondary metabolite production and plant recovery following the deep-cut treatment, whereas the reserves in the shallow-cut treatment were sufficient to provide for both vegetative growth and essential oil biosynthesis.

Although there was no significant interactive effect, there was a significant difference in cumulative essential oil yield for control and shallow-cut treatments in spring. The cumulative essential oil yield from a single harvest of the control at the end of the trial produced 17.05 ± 1.21 g DW/m^2^. The material originally harvested when shallow-cut treatment was implemented produced 3.11 ± 0.15 g DW/m^2^ at zero time (accumulated from April 2018 to December 2018), which, along with the final harvest, had a combined production of 14.90 ± 0.48 g DW/m^2^, which is 14% lower than the cumulative yield in the control over the same period. On this basis, the data might recommend biennial harvests, however, the continuation of the trial into a second year is likely to have shown an increase in woody material in the control, together with a reduced number of laterals. Pruning of *Kunzea*, such as that implemented with the shallow-cut treatment, should improve the growth of laterals and lower branches, providing for a spreading plant structure that intercepts more light and would provide for a higher oil yield [44,45,46]. This was evidenced in the coppiced stands of *Eucalypts globulus*, where a 40% loss of foliage resulted in the diversion of resources to the development of a higher leaf area, within and above, the defoliated crown zone [30,47]. Further studies in *Kunzea* examining the effect of harvest intensity on plant structure over a period of years are warranted.

Seasonal changes can influence oil yield [48]. However, we found no seasonal variation in the oil content (% DW) of *K. ambigua*. Simmons and Parsons [49] found that the essential oil extracted from *Eucalyptus ovata* (*Myrtaceae*), another Australian native species that occupies a similar ecological niche to *Kunzea*, showed relatively constant oil yield throughout the year. The findings reported here are consistent with previous studies of oils in *Myrtaceae* [50,51], and this is perhaps caused by relatively constant ecological pressures in these perennial, evergreen species.

### 3.3. The Effect of Harvest Intensity and Season on the Quality of Kunzea Essential Oil

There was a significant interaction effect of harvest intensity and season on α-pinene and bicyclogermacrene, which were significantly higher in oils from shallow- and deep-cut treatments harvested in spring compared to control. Terpenoids are important for plant survival, and several terpenoids have their roles in plant defense against biotic and abiotic stress [19]. Defoliation can trigger plant defense mechanisms including changing the chemical profile due to the expression of different genes [18]. Enhanced levels of α-pinene and bicyclogermacrene in shallow- and deep-cut treatments could be the result of plant response to damage to vegetative structures. α-Pinene exhibits diverse biological activity and have antiviral, antimicrobial, and antibacterial properties towing to their toxic effects on membranes, but there is limited data on the anti-herbivore properties of this monoterpene [52,53]. The production of sesquiterpenes, such as bicyclogermacrene, is associated with a defense response against pathogens and herbivores in members of the families *Solanaceae*, *Vitaceae* and *Meliaceae* [20]. Further, Durán-Peña et al. [54] found that bicyclogermacrene showed cytotoxic activity. It is possible that abiotic stress can be managed as an agronomic strategy for improving the quality of horticultural produce through enhanced levels of bioactive constituents such as bicyclogermacrene and α-pinene [21]. In addition to the elevated levels of α-pinene and bicyclogermacrene associated with recovery following defoliation, enhanced levels of bioactive constituents such as 1,8-cineole and viridiflorol were also found to coincide with autumn harvest, irrespective of harvest intensity. Owing to these components, kunzea oil has been shown to exhibit several therapeutic properties, such as anti-inflammatory activity, fumigant toxicity and anti-bacterial activity [55,56,57]. Terpinen-4-ol has also been shown to confer strong antimicrobial activity [58], though this study revealed that levels of this component were constant in *K. ambigua*, irrespective of season.

*K. ambigua* bears flowers between September and early November in Tasmania. This study showed a decrease in bioactive compounds, such as ledol and viridiflorol, during the spring flowering season. This aligns with the reported decrease in linalool and limonene in the essential oil of Thyme (*Thymus pulegioides* L.), which was attributed to the phenological stage of flowering [59].

### 3.4. Optimised Commercial Harvest Production of Kunzea Essential Oil

Harvesting time is species-specific and depends on the most ideal combination of chemical composition and yield, from a commercial point of view [60]. In addition, manipulating harvest dates and harvest intensity to maximise targeted oil profiles can be problematic as a variety of factors are involved in the chemical composition of essential oils rather than a single factor [61]. However, this study has shown that undertaking shallow-cut harvests in autumn could be conducive to maximising the oil quantity, producing premium kunzea oil with enhanced levels of bioactive constituents, such as α-pinene, bicyclogermacrene 1,8-cineole and viridiflorol. At the conclusion of this field trial, the height of the un-cut *kunzea* plants (control) rendered the continuation of this treatment unmanageable, whereas the poor recovery of deep-cut plants made the implementation of this harvest method unviable over two seasons. Shallow-cut harvest presented as the only commercially feasible management strategy; however, further study is needed to determine the long-term response to shallow-cut harvests in terms of changing chemical profiles with plant age, replenishment of soil nutrition and irrigation.

## 4. Materials and Methods

### 4.1. Materials

Analytical grade ethanol was purchased from Sigma-Aldrich, St Louis, USA. The internal standards, octadecane and C_7_–C_40_ saturated alkanes standard mix (Lot #LRAC3115) were also sourced from Sigma-Aldrich, St Louis, MO, USA. The total starch assay kit was purchased from Megazyme Pty Ltd., Warriewood, NSW, Australia, Cat. #K-TSTA. All incidental chemicals and reagents used were of analytical grade.

### 4.2. Experimental Location, Climate and Soil Characteristics

The trial was conducted at a commercial kunzea farm (latitude: −41.034816, longitude: 147.560118) in North-East Tasmania (Pioneer), Australia. The climatic characteristics of the trial site were obtained from SILO weather extrapolation (Figure 3). The SILO database and tool is a comprehensive archive of climate data recorded by the Australian Government Bureau of Meteorology [62]. The top-soil was analysed by CSBP, Bibra Lake, Australia (Table 5). There was no irrigation at the trial site during the study.

### 4.3. Plant Materials and Experimental Design

The trial site had been commercially harvested in April 2018. Harvest intensity treatments were imposed on 28 December 2018 in a Randomized Complete Block Design (RCBD) of four blocks, each of which was subject to three harvest intensity treatments, randomly allocated within each block (approximately 0.06 ha in total). Treatments were: control (no harvesting), shallow-cut (0.2 m above ground) and deep-cut (0.1 m above ground). The above-ground heights for each block were recorded and randomly selected 1 m^2^ areas were sampled from each treatment on 22 February 2019 (‘Late Summer’), 9 April 2019 (‘Autumn’) and 6 December 2019 (‘Spring’). The sampled plants were cut and labelled in the field before being stored on ice in a thermal-resistant container for transport to the laboratory. The samples were then weighed (biomass), sub-samples were dried at 70 °C for 24 h to determine dry weights (DW) and the remainder were packed in plastic bags and stored at −18 °C prior to being extracted by steam distillation. Species identification was confirmed by Tasmanian Herbarium, Hobart, Tasmania. The Growth Rate (GR) of biomass at each sampling date was calculated using the difference in the above-ground biomass (DW/m^2^) recorded for each sampling date using the formula:GR of biomass (g DW per m^2^/day) = change in biomass (DW)/number of days

### 4.4. Isolation of Oils

Kunzea oil was obtained by steam distillation of leaves and twigs from wild *K. ambigua*. Each sample from each of the blocks at each seasonal time point was divided into 2 × 200 g duplicates and the essential oils were obtained by steam distillation for five hours using a Clevenger-type apparatus. Essential oils were weighed and calculated as grams of oil per gram of dry plant material (oil content: g/g DW). The oils were stored at 4 °C prior to gas chromatography (GC) analyses. Essential oil yield (g DW/m^2^) was calculated using the formula:Essential oil yield (g DW/m^2^) = (oil distilled (g))/(plant material (DW g)) × total DW g material harvest from 1 m^2^

Cumulative essential oil yield (g DW/m^2^) at each sampling date was calculated by adding the essential oil yield (g DW/m^2^) recorded at the beginning of the trial (28 December 2018) to that obtained from harvest at any given date. The major components of kunzea essential oil were α-pinene, 1,8-cineole, terpinen-4-ol, bicyclogermacrene, globulol, viridiflorol and ledol, as analysed by GC/FID (Section 4.5. Gas Chromatography) and are presented as a percentage of the total oil yield. All other chemical constituents were grouped, and their percentages were summed and reported as minor components.

### 4.5. Gas Chromatography

Quantitative analyses were performed by GC analysis of the kunzea essential oil using a Hewlett Packard, USA 5890 series ll gas chromatograph equipped with a flame ionization detector (FID) and HP-1 crosslinked methyl siloxane column (30 mm × 0.32 mm, film thickness 0.25 µm). Injector and detector temperatures were set at 220 °C and 300 °C, respectively. Oven temperature was increased from 60 °C to 210 °C at 6 °C/min and to 280 °C at 25 °C/min. Total analysing time was 28 mins. Octadecane was used as the internal standard. Samples were injected (2 µL) at a split ratio of 50:1. Peak areas and retention times were measured by electronic integration, and quantitation was obtained by peak normalization of GC-FID data. The amount of each component in the kunzea essential oil was calculated assuming a 1:1 response ratio of the component to the internal standard;
mg of component in kunzea essential oil = (area of peak)/(area of ocatadecane) × mg of octadecane

The percentage of chemical components in kunzea essential oil was calculated as follows:% in component in kunzea essential oil =
(mg of component in kunzea essential oil)/(mg of kunzea essential oil) × 100

Qualitative analyses of the kunzea oil were carried out by GC-mass spectrometry (MS) using a Brucker-300 triple quadrupole benchtop GC-MS (Bruker Corporation, Billerica, MA, USA). The same column was used with similar experimental conditions to those described above for GC-FID except that the carrier gas was helium at a flow rate of 1.2 mL/min. The ion source temperature was 220 °C and the transfer line was held at 290 °C. The range from m/z 35 to 350 was scanned three times every second. The identification of individual peaks was done using their Kovats Indices (KI) and mass spectral data were compared to those for standard compounds in the MS database of National Institute of Standards and Technology (NIST) [63,64] (Table 6).

### 4.6. Extraction Methodology for Soluble Sugars and Starch

For the determination of NSCs, a single, randomly selected plant was dug out from each treatment block at the end of the trial (6 December 2019). Leaves, branches, and roots were separated and washed free of adhering soil. The samples were dried at 70 °C for 10 days. Samples were then ground coarsely with a Wiley Mill. Soluble sugars were extracted from 100 mg of the dried powdered sample tissue using 3 mL of 80% (*v*/*v*) ethanol and samples were incubated at 60 °C for approximately 10 min. The extracts were centrifuged at 4000× *g* for 10 min at 8 °C. The supernatant was transferred to separate tubes. The pellet was extracted twice more as described, and the supernatants were combined and frozen until analysis for soluble sugars (fructose, glucose and sucrose) using High-Performance Liquid Chromatography (HPLC)-MS, and the pellet was then used for starch analysis.

### 4.7. Data and Statistical Analysis

Data are represented as means ± SD (*n* = 4). Statistical analyses were performed using SAS software (Version 9.4, SAS Institute INC., Cary, NC, USA). Cumulative above-ground biomass (g DW/m^2^), GR of biomass (g DW/m^2^/day), oil content (% DW), essential oil yield (g DW/m^2^), cumulative essential oil yield (g DW/m^2^) and the concentration of chemical constituents of kunzea essential oil (%) were subjected to two-way (harvest intensity × season) analysis of variance (ANOVA) followed by post-hoc Duncan’s multiple range tests. NSCs were analysed using ANOVA followed by post-hoc Duncan’s multiple range tests.

## 5. Conclusions

Harvesting *kunzea* plants by shallow- and deep-cut treatments mobilised NSCs to provide TSS for recovery, and the deficit in NSCs continued for up to 12 months after treatment. In particular, deep-cut treatment resulted in the plants having inadequate resources for re-growth and the production of essential oil. Un-cut plants (control) grew most quickly in summer, showing a GR of biomass three times higher relative to that generated over winter/spring. This is most likely caused by dormancy during winter and the diversion of resources to the production of flower buds during spring. Overall, shallow-cut treatment resulted in a sink strength sufficient for both vegetative growth and essential oil biosynthesis. An interactive effect of harvest intensity with season was found for α-pinene and bicyclogermacrene, whereby these components were elevated in both shallow- and deep-cut treatments in spring, relative to the control, possibly owing to the plant initiating protective mechanisms for the new growth following defoliation/harvest. Hence, in addition to mobilising resources, harvesting *K. ambigua* may increase the level of bioactive components such as α-pinene and bicyclogermacrene. In relation to seasonal variation, autumn harvests produced the highest percentage of other bioactive constituents such as 1,8-cineole and viridiflorol, though the practice of blending by industry would likely be required to ensure consistency of the product.

## Figures and Tables

**Figure 1 plants-12-00020-f001:**
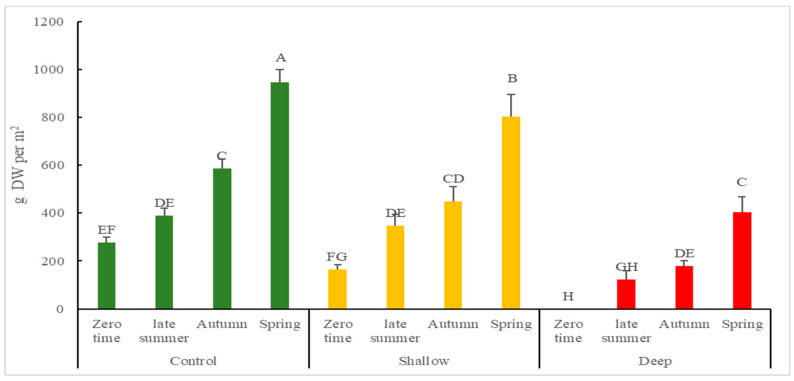
Cumulative above ground biomass (g DW/m^2^) of *K. ambigua* following harvest treatments of control, shallow-cut and deep-cut imposed in Summer (28 December 2018) and from sampling of simulated harvest of 1 m^2^ plots during subsequent late Summer (22 February 2019), Autumn (9 April 2019) and Spring (6 December 2019). Values are presented as means ± SD (*n* = 4). Mean values designated by a different letter are significantly different (*p* < 0.05) between groups according to two-way ANOVA test with initial harvest intensity (HI) and season (S) as variability factors. The zero-time data is existing biomass after last commercial harvest (April 2018).

**Figure 2 plants-12-00020-f002:**
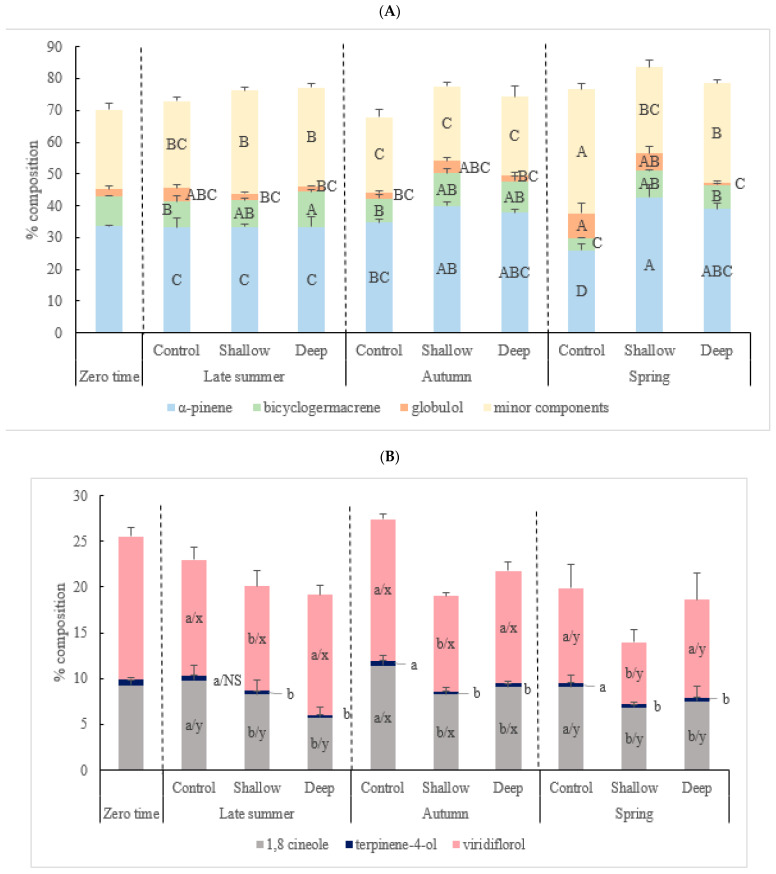
The significant interaction effect (**A**) and main effect of harvest intensity and season, respectively, (**B**) on the major chemical constituents (%) in kunzea oil. Values are presented as means ± SD (n = 4). Mean values designated by a different letter are significantly different between groups according to two-way ANOVA test with initial harvest intensity (HI) and season (S) as variability factors: initial harvest intensity (HI) for a, b, c and season (S) for x, y, and z using lowercase letters, and initial harvest intensity (HI) × season (S) interaction for capital letters. The zero-time data was not included in two-way ANOVA test. NS: not significant.

**Figure 3 plants-12-00020-f003:**
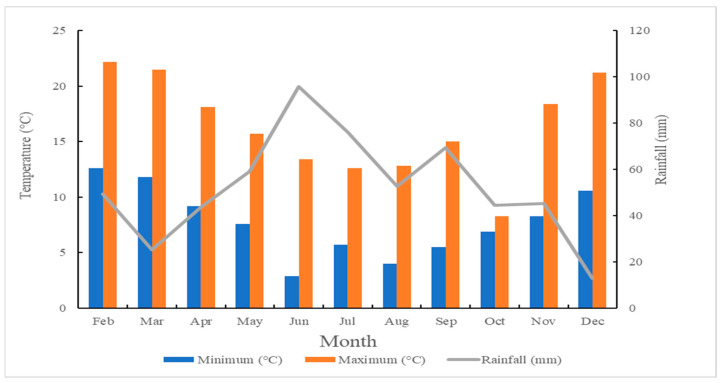
The mean monthly rainfall and minimum and maximum temperature at Pioneer, Tasmania from February to December in 2019 (obtained from the SILO, https://www.longpaddock.qld.gov.au/silo/point-data/ accessed on 3 July 2020).

**Table 1 plants-12-00020-t001:** Growth Rates (GR) of *Kunzea* plants following treatments of control, shallow-cut and deep-cut harvests imposed in Summer (zero time: 28 December 2018) and from subsequent simulated harvest of 1 m^2^ plots in late Summer (22 February 2019), Autumn (9 April 2019) and Spring (6 December 2019).

Season (S)	Initial Harvest Intensity (HI)	GR of Biomass (g DW/m^2^/Day)
Zero time	Control	
	Shallow	
	Deep	
Late summer	Control	6.98 ± 0.54 A
	Shallow	3.29 ± 0.18 C
	Deep	2.21 ± 0.32 DE
Autumn	Control	5.74 ± 0.40 B
	Shallow	2.80 ± 0.11 CD
	Deep	1.76 ± 0.22 EF
Spring	Control	2.76 ± 0.16 CD
	Shallow	1.87 ± 0.12 EF
	Deep	1.18 ± 0.18 F
HI		***
S		***
HI × S		***

Values are presented as means ± SD (*n* = 4). Mean values designated by a different letter are significantly different between groups (*** *p* < 0.0001), according to two-way ANOVA test with initial harvest intensity (HI) and season (S) as variability factors: initial harvest intensity (HI) for a, b, c and season (S) for x, y and z using lowercase letters, and initial harvest intensity (HI) × season (S) interaction for capital letters. The zero-time data was not included in two-way ANOVA test.

**Table 2 plants-12-00020-t002:** Starch and soluble sugars in leaves, branches and roots sampled on 6 December 2019 subsequent to harvest intensity treatments of control, shallow cut (0.2 m above ground) and deep-cut (0.1 m above the ground), which were imposed on 28 December 2018.

	Starch (mg/g DW)	Fructose (mg/g DW)	Glucose (mg/g DW)	Sucrose (mg/g DW)	Total Soluble Sugar (mg/g DW)	NSCs (mg/g DW)
Leaves						
Control	2.51 ± 0.08 a	9.48 ± 1.12 b	7.68 ± 0.99 c	23.53 ± 0.46 b	40.69 ± 2.56 c	43.21 ± 2.55 c
Shallow	1.90 ± 0.14 b	10.96 ± 0.58 b	11.53 ± 0.82 b	41.54 ± 2.22 a	64.02 ± 1.02 b	65.92 ± 1.11 b
Deep	1.82 ± 0.04 b	13.98 ± 1.04 a	14.31 ± 0.76 a	44.96 ± 2.17 a	73.25 ± 3.71 a	75.07 ± 3.74 a
Branches						
Control	4.24 ± 0.34 NS	6.35 ± 0.50 a	7.34 ± 0.65 a	19.38 ± 2.09 a	33.07 ± 3.23 a	37.31 ± 3.56 a
Shallow	5.29 ± 0.65	3.61 ± 0.60 c	3.48 ± 0.59 c	13.35 ± 1.87 b	20.44 ± 3.05 b	25.73 ± 3.26 b
Deep	5.08 ± 0.25	5.06 ± 0.33 b	5.76 ± 0.59 b	19.14 ± 1.60 a	29.97 ± 1.50 a	35.05 ± 1.33 a
Roots						
Control	9.31 ± 0.89 a	3.62 ± 0.46 NS	3.83 ± 0.45 NS	16.79 ± 2.58 a	24.23 ± 3.45 a	33.54 ± 3.48 a
Shallow	8.33 ± 1.21 a	3.63 ± 0.56	3.50 ± 0.55	13.43 ± 1.73 b	20.56 ± 2.83 ab	28.89 ± 2.05 ab
Deep	7.51 ± 0.59 b	3.13 ± 0.34	3.01 ± 0.30	12.27 ± 0.35 b	18.41 ± 0.77 b	25.92 ± 0.96 b

Values are presented as means ± SD (*n* = 4). Mean values designated by a different letter are significantly different between treatments (*p* < 0.05). NS: not significant.

**Table 3 plants-12-00020-t003:** Oil content (% DW), essential oil yield (g DW per m^2^) and cumulative oil yield (g DW per m^2^; sum of biomass at time zero and that collected at subsequent harvest dates) of *K. ambigua* following treatments of control, shallow-cut and deep-cut harvests imposed in Summer (zero time: 28 December 2018) and from subsequent simulated harvest of 1 m^2^ plots in late Summer (22 February 2019), Autumn (9 April 2019) and Spring (6 December 2019).

Season (S)	Initial Harvest Intensity (HI)	Essential Oil Content (% DW)	Essential Oil Yield (g DW Per m^2^)	Cumulative Essential Oil Yield (g DW Per m^2^)
Zero time	Control	1.96 ± 0.24 a	7.01 ± 1.01 C	0 H
	Shallow	2.04 ± 0.12 a	3.11 ± 0.15 E	3.11 ± 0.15 G
	Deep	1.96 ± 0.24 b	0 F	7.01 ± 1.01 F
Late summer	Control	1.84 ± 0.06 a	7.19 ± 0.96 C	7.19 ± 0.96 EF
	Shallow	1.98 ± 0.20 a	3.65 ± 0.89 DE	6.76 ± 1.04 F
	Deep	1.66 ± 0.13 b	2.05 ± 0.53 EF	9.06 ± 1.54 DEF
Autumn	Control	1.90 ± 0.15 a	11.13 ± 1.31 B	11.13 ± 1.31 CD
	Shallow	2.00 ± 0.13 a	5.72 ± 0.31 CD	8.83 ± 0.46 DEF
	Deep	1.65 ± 0.13 b	2.97 ± 0.18 E	9.98 ± 1.09 DE
Spring	Control	1.80 ± 0.19 a	17.05 ± 1.21 A	17.05 ± 1.21 a/A
	Shallow	1.84 ± 0.11 a	11.79 ± 0.23 B	14.90 ± 0.48 b/AB
	Deep	1.54 ± 0.32 b	6.24 ± 0.18 C	13.25 ± 0.65 c/BC
HI		*	***	*
S		NS	***	***
HI × S		NS	**	***

Values are presented as means ± SD (*n* = 4). Mean values designated by a different letter are significantly different between groups (*** *p* < 0.0001, ** *p* < 0.001, * *p* < 0.05, NS = not significant), according to two-way ANOVA test with initial harvest intensity (HI) and season (S) as variability factors: initial harvest intensity (HI) for a, b, c and season (S) for x, y and z using lowercase letters, and initial harvest intensity (HI) × season (S) interaction for capital letters. The zero-time data was not included in two-way ANOVA test.

**Table 4 plants-12-00020-t004:** The major and minor chemical components in kunzea oil (%) from two-way ANOVA with initial harvest intensity (HI), season (S) and their interaction (HI × S) as variability factors.

	The Major Chemical Components in Kunzea Oil								Minor Components
	α-Pinene	1,8-Cineole	Terpinen-4-ol	α-Terpineol	Bicyclogermacrene	Globulol	Viridiflorol	Ledol	
HI	***	**	**	**	***	**	*	*	NS
S	*	*	NS	*	***	*	**	*	***
HI × S	***	NS	NS	NS	*	*	NS	NS	***

*** *p* < 0.0001, ** *p* < 0.001, * *p* < 0.05, NS = not significant (*p* > 0.05), and *n* = 4.

**Table 5 plants-12-00020-t005:** The physicochemical properties of topsoil to 200 mm depth, which represents the distribution of the majority of the root profile of *K. ambigua* in the field trial.

Texture	Nitrate Nitrogen (mg/kg)	Phosphorus (mg/kg)	Potassium (mg/kg)	Sulfur (mg/kg)	pH Level (Cacl_2_)	DTPA Copper (mg/kg)	DTPA Iron (mg/kg)	DTPA Manganese (mg/kg)
1.88 ± 0.25	<1	8.25 ± 1.50	67.25 ± 36.39	2.37 ± 0.47	4.25 ± 0.26	0.52 ± 0.15	156.78 ± 31.57	1.36 ± 0.98

Values are presented as means ± SD (*n* = 4).

**Table 6 plants-12-00020-t006:** Identified chemical components in kunzea oil ^1^ extracted from *K. ambigua*.

	Components	RT ^2^	KI ^3^ Calc.	KI Lit ^4^ Ref
1	α-pinene	2.906	917	938
2	camphene	3.401	969	952
3	sabinene	3.44	973	977
4	β-pinene	3.618	990	982
5	limonene	4.018	1015	1032
6	1,8-cineole	4.253	1026	1036
7	(Z)-B-ocimene	4.517	1038	1036
8	(E)-B-ocimene	4.706	1046	1046
9	isoamyl butyrate	4.889	1054	1054
10	cis-linalool oxide	5.428	1075	1071
11	terpinolene	5.53	1078	1085
12	linalool	5.938	1093	1097
13	isoamyl isovalerate	6.173	1102	1102
14	trans-pinocarveol	6.646	1142	1143
15	pinocarvone	6.719	1148	1165
16	terpinen-4-ol	6.925	1165	1170
17	α-terpineol	7.194	1186	1197
18	citronellol	7.958	1228	1227
19	geraniol	8.493	1253	1252
20	ϒ-elemene	9.695	1303	1333
21	α-cubebene	10.531	1351	1352
22	α-copaene	10.972	1374	1380
23	β-elemene	11.299	1391	1392
24	α-gurjunene	11.641	1409	1412
25	β-caryophyllene	11.828	1419	1425
26	aromadendrene	12.228	1441	1444
27	a-humulene	12.415	1450	1461
28	allo-aromadendrene	12.645	1462	1466
29	germacrene D	13.189	1489	1468
30	bicyclogermacrene	13.36	1497	1501
31	calamenene	13.88	1525	1527
32	palustrol	14.699	156	1577
33	spathulenol	14.896	1578	1584
34	globulol	15.013	1584	1594
35	viridiflorol	15.189	1593	1603
36	ledol	15.375	1601	1613
37	isospathulenol	16.07	1622	1639
38	α-muurolol	16.308	1628	1654

^1^ Kunzea oil was extracted by steam distillation for five hours; ^2^ RT: Retention time; ^3^ KI: Kovats index; ^4^ KI ref: Thomas, et al. [64]; Values are presented as means ± SD (*n* = 4).

## Data Availability

Not applicable.

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
