# Peer review of "The Effect of the Height of Coppicing and Harvest Season on the Yield and Quality of the Essential Oil of Kunzea ambigua"

_plants, 2022, doi:10.3390/plants12010020_

Round 1

Reviewer 1 Report

In general, this study was organized and conducted well. In the introduction, it would be nice to add much more information about this plant and its importance for traditional medicine, the use in everyday life of the natives, the breadth of the use of essential oil in various industries, medicine and cosmetics. To reveal the importance and value of this plant for the region, its economic importance. The electronic version of the journal allows you to add a few more major publications on the research topic.

That pruning will affect the yield of essential oil is understandable, but did the authors try to use fertilizers? Organic? Mineral? complex with trace elements? The quality of the writing of plants, the availability of their basic nutrients will greatly affect the yield and component composition of essential oils.

If this moment (assessment of the effect of fertilizers on the yield and quality of essential oil) has not yet been taken into account by the authors, then I would like to hope that in the very near future they will take this into account and significantly improve their research.

It is important to clarify: essential oils were distilled only from the leaves? from leaves and flowers? from branches, leaves and flowers? Just one time? It is important to show whether there is a dynamics of changes in the qualitative and quantitative composition of essential oils for each organ (or the traditional raw materials used for a given species).

Numerical data in the article should be given in rounded form - up to tenths, and not up to hundredths.

For example: not 5.29 ± 0.65 but 5.3 ± 0.7; not 3.61 ± 0.60 but 3.6 ± 0.6 and so on, since the reduction of digital values \u200b\u200bwith hundredths in this case is of no fundamental importance.

Author Response

Response to Reviewers’ comments: Thank you very much for your consideration, and we really appreciate the comments that have resulted in an improved manuscript. Changes have been made and are highlighted (red) in the revised manuscript according to the suggestions of reviewers.

Reviewer 1)

Comments and suggestions for authors: In general, this study was organized and conducted well. In the introduction, it would be nice to add much more information about this plant and its importance for traditional medicine, the use in everyday life of the natives, the breadth of the use of essential oil in various industries, medicine and cosmetics. To reveal the importance and value of this plant for the region, its economic importance. The electronic version of the journal allows you to add a few more major publications on the research topic.

Response: Currently, there is limited research on kunzea oil and kunzea ambigua. Therefore, we include the content based on the existing references. As shown in <Introduction>, there is information about the native habitat and plant growth of Kunzea ambigua together with essential oils (colour, fragrance, the use of essential oils). We add the information about current commercial products with kunzea oil (Row 37-39: Currently, commercial products with kunzea oil have many uses in cosmetics and personal care products such as oil balm, body spray and soap.)

That pruning will affect the yield of essential oil is understandable, but did the authors try to use fertilizers? Organic? Mineral? complex with trace elements? The quality of the writing of plants, the availability of their basic nutrients will greatly affect the yield and component composition of essential oils. If this moment (assessment of the effect of fertilizers on the yield and quality of essential oil) has not yet been taken into account by the authors, then I would like to hope that in the very near future they will take this into account and significantly improve their research.

Response: In this study, the kunzea commercial plantation is under organic farming. In particular, the kunzea farmer wants to maintain certified organic kunzea oils for his products. So, we respect his thought together with his business. Therefore, we set the plant recovery trial under organic management in response to the aspirations of our industry partner to optimise the commercial production of kunzea oil. However, we recognised the importance of further research (the effect of fertilisers on the yield and quality of essential oils).

It is important to clarify: essential oils were distilled only from the leaves? from leaves and flowers? from branches, leaves and flowers? Just one time? It is important to show whether there is a dynamics of changes in the qualitative and quantitative composition of essential oils for each organ (or the traditional raw materials used for a given species).

Response: We already mentioned the specific conditions for steam distillation (5 hrs steam distillation) in Materials & Methods section (4.4 Isolation of oils_row361-363). We add the information of the distillation sample (twigs and leaves) in row 359-360..

Numerical data in the article should be given in rounded form - up to tenths, and not up to hundredths. For example: not 5.29 ± 0.65 but 5.3 ± 0.7; not 3.61 ± 0.60 but 3.6 ± 0.6 and so on, since the reduction of digital values \u200b\u200bwith hundredths in this case is of no fundamental importance.

Response: Thank you for the suggestion. We checked the style of numerical data in recently published paper from MDPI-plants (Plants | Topical Collection : Essential Oils of Plants (Chemical Composition, Variation and Properties) (mdpi.com)). The numerical data format in recently published papers is the same as our data. Thus, authors decide to keep our numerical data. Could you please consider our opinions?

  1. Effects of Essential Oil Fumigation on Potato Sprouting at Room-Temperature Storage

https://doi.org/10.3390/plants11223109

  1. Chemical Composition, Antioxidant, In Vitro and In Situ Antimicrobial, Antibiofilm, and Anti-Insect Activity of Cedar atlantica Essential Oil

https://doi.org/10.3390/plants11030358

  1. Evaluation of Essential Oils as Sprout Suppressants for Potato

(Solanum tuberosum) at Room Temperature Storage

https://doi.org/10.3390/plants11223055

Reviewer 2 Report

The topic of the paper „The effect of the height of coppicing and harvest season on the yield and quality of the essential oil of Kunzea ambigua” is very interesting for readers.

The aim of this study was to compared the regrowth, essential oil content and composition of kunzea plants after harvesting vegetative material to a depth of 0.2 m above ground level (shallow-cut), relative to plants cut to a depth of 0.1m above ground level (deep-cut) over the 2018/2019 growing season.

The experimental results show that total soluble sugar concentrations were higher in the leaves and lower in the roots of deep-cut treated plants compared to the other treatments, indicating defoliated K. ambigua responds by mobilizing sugars into above-ground biomass.

The introduction provides sufficient background and includes relevant references.

The design research is well described.

The manuscript is well written, and the text is easy to read.

The results are consistent and clearly presented.

At the reference list, some names of species are not italic:

-         at reference number 8: Gliricidia sepium, Leucaena leucocephala

-         at reference number 14: Erythrina poeppigiana, Gliricidia sepium

-         at reference number 16, 17, 36: Camellia sinensis L.

-         at reference number 29: Leymus chinensis

-         at reference number 30: Eucalyptus globulus

-         at reference number 43: Cymbopogon flexuosus

-         at reference number 52: Pistacia vera L.

-         at reference number 59: Thymus pulegioides L.

Author Response

Reviewer 2)

Response to Reviewers’ comments: Thank you very much for your consideration, and we really appreciate the comments that have resulted in an improved manuscript. Changes have been made and are highlighted (red) in the revised manuscript according to the suggestions of reviewers.

Comments and suggestions for authors: The topic of the paper „The effect of the height of coppicing and harvest season on the yield and quality of the essential oil of Kunzea ambigua” is very interesting for readers.

The aim of this study was to compared the regrowth, essential oil content and composition of kunzea plants after harvesting vegetative material to a depth of 0.2 m above ground level (shallow-cut), relative to plants cut to a depth of 0.1m above ground level (deep-cut) over the 2018/2019 growing season.

The experimental results show that total soluble sugar concentrations were higher in the leaves and lower in the roots of deep-cut treated plants compared to the other treatments, indicating defoliated K. ambigua responds by mobilizing sugars into above-ground biomass.

The introduction provides sufficient background and includes relevant references.

The design research is well described.

The manuscript is well written, and the text is easy to read.

The results are consistent and clearly presented.

 At the reference list, some names of species are not italic:

-         at reference number 8: Gliricidia sepium, Leucaena leucocephala

-         at reference number 14: Erythrina poeppigiana, Gliricidia sepium

-         at reference number 16, 17, 36: Camellia sinensis L.

-         at reference number 29: Leymus chinensis

-         at reference number 30: Eucalyptus globulus

-         at reference number 43: Cymbopogon flexuosus

-         at reference number 52: Pistacia vera L.

-         at reference number 59: Thymus pulegioides L.

Response: We revised the format which you mentioned in Reference.

Reviewer 3 Report

Manuscript title: The effect of the height of coppicing and harvest season on the yield and quality of the essential oil of Kunzea ambigua

Manuscript ID:  plants-2099566

Journal:   Plants

The aim of this study was to determine the influence of harvest intensity and season on biomass accumulation, oil yield, and composition of kunzea oil. Also, the level of NSCs in the leaves, stems and roots were measured to inform how K. ambigua mobilises resources after partial defoliation due to harvest. The idea is sound and the manuscript is well written. Material and methods contain details which help other researchers to follow easily. Statistical analysis was well performed. Results and discussion are correlated and updated. Academic English language is fine. However, some minor suggestions need to be considered before accepting the current version such as:

Replace material and methods before the results section;

Discussion should be re-written and the authors are asked to make it one unit without subtitles.

Author Response

Response to Reviewers’ comments: Thank you very much for your consideration, and we really appreciate the comments that have resulted in an improved manuscript. Changes have been made and are highlighted (red) in the revised manuscript according to the suggestions of reviewers.

Reviewer 3)

Comments and suggestions for authors: The aim of this study was to determine the influence of harvest intensity and season on biomass accumulation, oil yield, and composition of kunzea oil. Also, the level of NSCs in the leaves, stems and roots were measured to inform how K. ambigua mobilises resources after partial defoliation due to harvest. The idea is sound and the manuscript is well written. Material and methods contain details which help other researchers to follow easily. Statistical analysis was well performed. Results and discussion are correlated and updated. Academic English language is fine. However, some minor suggestions need to be considered before accepting the current version such as:

Replace material and methods before the results section;

Response: According to MDPI plants template, Results section comes first. We followed the template under journal instructions (1. Introduction / 2. Results / 3. Materials and Methods / 4. Discussion / 5. Conclusions).

Discussion should be re-written and the authors are asked to make it one unit without subtitles.

Response: We wrote the Discussion along with the Results. We thought that it is logical to expand our thoughts according to subtitles from Results. Also, the reader would be easier to understand the manuscript.

<Results> 2.1. Growth rates and above-ground biomass following harvest of kunzea plants according to harvest intensity and season

2.2. The levels of starch, soluble sugars, and NSCs in coppiced kunzea plants influencing on essential oils biosynthesis

2.3. The effect of harvest intensity and season on the chemical constituents of kunzea essential oil (%)

<Discussions>

3.1. Plant growth and the allocation of NSCs of K. ambigua in response to harvest intensity and season

3.2. The effect of harvest intensity and season on the quantity of kunzea essential oil

3.3. The effect of harvest intensity and season on the quality of kunzea essential oil

3.4. Optimised commercial harvest production of kunzea essential oil
